# Perspectives for Uses of Propolis in Therapy against Infectious Diseases

**DOI:** 10.3390/molecules27144594

**Published:** 2022-07-19

**Authors:** Antonio Salatino

**Affiliations:** Institute of Biosciences, Department of Botany, University of São Paulo, Rua do Matão 277, São Paulo 05508-090, SP, Brazil; asalatin@ib.usp.br

**Keywords:** CAPE, COVID-19, flavonoids, microbial pathogens, plant resins, *Staphylococcus aureus*

## Abstract

Propolis has gained wide popularity over the last decades in several parts of the world. In parallel, the literature about propolis composition and biological properties increased markedly. A great number of papers have demonstrated that propolis from different parts of the world is composed mainly of phenolic substances, frequently flavonoids, derived from plant resins. Propolis has a relevant role in increasing the social immunity of bee hives. Experimental evidence indicates that propolis and its components have activity against bacteria, fungi, and viruses. Mechanisms of action on bacteria, fungi, and viruses are known for several propolis components. Experiments have shown that propolis may act synergistically with antibiotics, antifungals, and antivirus drugs, permitting the administration of lower doses of drugs and higher antimicrobial effects. The current trend of growing resistance of microbial pathogens to the available drugs has encouraged the introduction of propolis in therapy against infectious diseases. Because propolis composition is widely variable, standardized propolis extracts have been produced. Successful clinical trials have included propolis extracts as medicine in dentistry and as an adjuvant in the treatment of patients against COVID-19. Present world health conditions encourage initiatives toward the spread of the niche of propolis, not only as traditional and alternative medicine but also as a relevant protagonist in anti-infectious therapy. Production of propolis and other apiary products is environmentally friendly and may contribute to alleviating the current crisis of the decline of bee populations. Propolis production has had social-economic relevance in Brazil, providing benefits to underprivileged people.

## 1. Introduction

“Healing with medicinal plants is as old as mankind itself” [1]. During most of the Paleolithic period, the use of plants for medicinal purposes probably was instinctive, as also has been with animals. Archeological findings have shown that plants were deliberately used to treat health problems [2]. In present day, human groups in traditional societies, for example, in New Guinea, still rely on pharmaceutical knowledge based on hundreds of species from 99 plant families, which have been used deliberately for medicinal purposes [3].

Despite the advancements in chemical synthesis and its application in the production of synthetic drugs, a considerable proportion of substances used to date in modern medicine are plant derived. It is estimated as 35% proportion of modern medicines originated directly or indirectly from natural products, most of them from plants (25%) and microorganisms (13%).

Propolis (or bee glue) is a sticky product with a resinous flavor, used by honeybees to repair damages in the nest, avoid abrupt changes in temperature in the nest, and protect the hive against predators and microorganisms. It is a complex product containing resins collected from plants, which are mixed with beeswax and salivary enzymes [4]. Medicinal uses of propolis can be traced back to 13,000 BCE [5]. A decline in the medicinal use of propolis occurred during the last century, concomitant with the rise of the use of drugs in pure form (either synthetic or natural) for industrial production of medicines. Therapeutical uses of propolis survived in the last century in Balkan countries and Soviet clinics, including treatment of tuberculosis [6]. After World War II, a revival in the medicinal interest of natural resources ensued in Western countries. Currently, propolis is used as a medicine in some countries (e.g., Germany) or as a complementary food, in addition to a popular and alternative medicine in several others. The gradual revival of the interest in propolis encouraged research on its medicinal properties, beginning with antibacterial effects. Over the past decades, propolis research has uncovered a wide diversity of biological effects, including anesthetic, antiaging, antibacterial, anticarcinogenic, antidiabetic, antifungal, anti-hypertensive, anti-inflammatory, antioxidant, antiprotozoal, antiulcer, antiviral, wound-healing, cytotoxic, hepatoprotective, and immunomodulatory [4,7,8,9,10].

Several reviews have been published recently, dealing with the antimicrobial properties of propolis [11,12]. The present review aims to advance a step forward relative to the available literature by including discussions about the benefits of the introduction of propolis in therapy against infectious diseases, considering the adverse effects of the currently used drugs, and the escalating trend of development of antidrug microbial resistance. In addition, the present review discusses measures that have been taken toward the production of standardized propolis extracts, attempts to obtain highly active fractions from crude propolis extracts, clinical uses of propolis in dentistry and hospital therapies, as well as environmental and social-economic benefits derived from the production of propolis and other apiary products.

## 2. Propolis Composition

Propolis composition is highly variable, depending on several factors, such as the geographic location of the hives, season of the year, and characteristics of the local vegetation [13]. Such variability represents a challenge for standardization and quality control of propolis [14]. In addition to plant resins and beeswax, propolis contains pollen [15]. It is common in propolis literature quotations, varying little one from the other, according to which propolis typically contains “30% beeswax, 55% vegetal resins and balsams, 10% volatile oil and around 5% pollen”. It seems that these data have no experimental basis. An attempt to trace their origin in the literature had no success [16]. In fact, these quotations are misleading; no quantification has ever been performed, for example, about the contents of pollen in propolis. Moreover, the content of pollen in propolis, around 5% (some authors admit values up to 10%), is an over-exaggeration, because pollen is a propolis contaminant from wind-carried grains [17], and pollen adhered to the bodies of the laborer bees. Propolis has no “typical” composition; on the contrary, it is well known to vary widely from one type to another and from region to region [18]. Propolis of the same type may present widely distinct contents of total phenolic components. An inverse proportion of the contents of triterpenoids versus phenolic substances is observed in green propolis derived from *Baccharis dracunculifolia* in Bolivia and Brazil [19,20]. Propolis containing high phenolic contents are likely to have high antioxidant and antibacterial activity, in contrast with propolis of the same type with high contents of triterpenoids and beeswax and low amounts of phenolic components. Major constituents of propolis are beeswax, whose content rarely reaches 30%, which is the upper limit admitted for Brazilian green propolis [21]. Propolis wax contains mostly hydrocarbons, esters, fatty acids, and alcohols [22]. Another major propolis component containing virtually all the substances with biological activity in propolis is the resin collected from plants. Several reviews have been published about types of propolis, respective plant sources of resin, composition, and biological activities (e.g., [9]). Propolis resin has a complex composition, comprising over 800 components, considering propolis types from all geographic regions [14]. The main components of most propolis types are phenolic compounds. Two types of propolis stand out for having been the object of numerous studies: temperate propolis from both new and old worlds, derived from bud exudates of poplar trees (*Populus nigra* and related species; Salicaceae), and Brazilian green propolis, derived from the resin obtained from vegetative buds and young leaf primordia of “alecrim-do-campo” (*Baccharis dracunculifolia*; Asteraceae). In Figure 1 are shown structures of constituents of several types of propolis, as follows: (a) caffeic acid phenethyl ester (CAPE; **I**) and the flavonol galangin (**II**), both from poplar or temperate propolis; (b) the prenylated phenylpropanoid artepillin C (**III**), the flavonol kaempferide (**IV**), and the diterpenoid agathic acid (**V**), components of Brazilian green propolis; (c) nemorosone (**VI**), a polyisoprenylated benzophenone of propolis from Amazonian [23], Cuban and Venezuelan propolis, derived from species of *Clusia* (Clusiaceae; [24]; (d) the isoflavonoid medicarpin (VII) from Brazilian red propolis, derived from *Dalbergia ecastaphyllum* (Leguminosae, Faboideae) [25]; (e) the prenylated flavanone propolin A (or nympheol A; VIII), from Taiwanese and Okinawan propolis, derived from *Macaranga tanarius* (Euphorbiaceae) [26]; (f) a stilbene with prenyl-substituents on ring A (**IX**) from Australian propolis, derived from *Lepidosperma* sp. (Cyperaceae) [27]; (e) a stilbene with prenyl-substituents on both A and B rings (e.g., **X**) from Ghanan propolis [28]; (f) epigallocatechin-gallate (**XI**), a flavanonol glycoside from Turkish propolis [29]; (g) a podophyllotoxin derivative, from a stingless bee propolis from Indonesia. In Figure 2 are presented examples of propolis types with known plant sources of resin and respective world location. Propolis has high stability over time: fresh and aged propolis (up to 180 days old) exhibit similar chemical composition and antibacterial activity [30]. This is probably a consequence of the high chemical stability of two major classes of propolis constituents: beeswax and phenolic substances.

The combined or synergistic effect of propolis components, among which are phenylpropanoids, phenolic esters, and flavonoids, is the actual determinant of the biological activities of propolis [34,35,36,37,38,39]. A challenging aspect involving propolis activity is the determination of which among its components are agents likely responsible for the observed biological effects. Regarding antibacterial activity, flavonoids have been recognized as relevant active components. Among flavonoid classes, chalcones possess higher antibacterial activity, followed by flavanes and flavan-3-ols. Structural details, such as hydroxylation and prenylation, may enhance flavonoid antibacterial efficiency [40], an observation in line with the commented antibacterial activity of propolins. Some chalcones have antibacterial activity considerably stronger than antibiotics in current use. For example, the MIC value against *S. aureus* of the chalcone isobavachalcone, a component of *Dorstenia barteri* (Moraceae), was 0.33 [40]. This value is impressive if compared with MIC values of current antibiotics, such as: amoxicillin: 2; ampicillin: 4; penicillin: 0.06; enrofloxacin: 0.25; chloramphenicol: 4–8; trimethoprim: 19 [41]. Chalcones are found more commonly in species of the Leguminosae family, followed by species of Moraceae and Asteraceae, although some species of other families, such as Annonaceae, Myrtaceae, Piperaceae, and Pteridaceae, have been reported to contain chalcones [42]. So far, chalcones have been reported from propolis types containing resins derived from Leguminosae plants, irrespective of the subfamily of the species. For example, chalcones occur in Argentinean propolis (derived from *Zuccagnia punctata*, subfamily Caesalpinioideae [43], as well as in green propolis from northeast Brazil (derived from *Mimosa tenuiflora*, subfamily Mimosoideae; [44]), Brazilian red propolis (derived from *Dalbergia ecastaphyllum*, subfamily Faboideae; [45]) and Nepalese propolis (probably derived from a *Dalbergia* species, subfamily Faboideae) [46]. Kaempferide (Figure 2, **IV**), a flavonol component of Brazilian green propolis, is used in the treatment of skin infections caused by *Staphylococcus aureus* [11]. Luteolin, a flavone commonly present in propolis, exhibits MIC (minimum inhibitory concentration) of 1.5 µg mL^−1^ against *S. aureus* [40]. Quercetin, the most common flavonoid in plants and often reported in propolis composition, has MIC values of 1.8 against *S. aureus* and 2.5 against *E. coli*. This flavonol is assumed to exert fractional bacterial lysis changes on the cell permeability, which influence the synthesis of nucleic acids and proteins, and reduces enzyme activities [47]. Propolis constituents that are inactive against bacteria, such as beeswax, contribute to adding volume and mass to the product while reducing MIC value. It has long been recognized in propolis an association between total phenols and total flavonoids as well as an association between high contents of phenols and flavonoids with high antioxidant and antibacterial activities [48]. The higher or lower content of flavonoids and other phenolic substances probably accounts for the observed variability of antibacterial activity of propolis samples of the same type (Table 1).

## 3. Plant Resins and Honeybee Immunity

Apitherapy has developed to the point of becoming a huge commercial market in many parts of the world. Abundant scientific evidence has been provided about the biological effects of honey, propolis, and royal jelly [67], with the potential of turning these substances useful for the treatment of infections, cancer, and other health problems of humans and domestic animals [68]. We are far, however, from witnessing the production of medicines containing propolis by the pharmaceutical industry. Many reasons explain why propolis remains a complementary and alternative medicine in most parts of the world. Certainly, a strong reason is uncertainty by researchers and authorities in medical sciences about real propolis medicinal effects. For this reason, an evaluation of the antibiotic role of propolis in honeybee biology rather than in human health may provide convincing data and may attenuate skepticism about the medicinal usefulness of propolis. This is an approach in line with biomimicry, an area of science with a focus on nature’s models and on what we can learn from them [69]. It is pertinent in this context that the *Apis* innate immune system contains far fewer immunity-related genes than the immune system of other insects [70]. The reduced number of immune genes is compensated in honeybees by a well-developed behavioral defense mechanism of ‘social’ or ‘collective immunity’ [71]. Self-medication by honeybees was first proposed by Simone-Finstrom and Spivak [72]. They demonstrated that honeybee colonies are stimulated to increase the collection of resins in response to a fungal infection. In addition, a decrease in infectious diseases was noted in colonies experimentally enriched with plant resins. Later, Spivak et al. [73] argued that the use of plant resins by honeybees cannot be compared to the self-medication and pharmacophagy exerted by several insects, such as moths, fruit flies, and butterflies, which consume plants loaded with toxins [74,75,76]. In these cases, self-medication benefits individual organisms. For this reason, the term social medication has been proposed [73]. Instead of pharmacophagy, the collection of plant resins and the subsequent storage of propolis by honey bees is a case of pharmacophory, comparable to the incorporation by birds of fresh aromatic plants into their nests; such behavior is interpreted as an evolutionary adaptation against bird ectoparasites and pathogens [77]. A curious case of pharmacophory in urban environments involves the incorporation into bird nests of butts from smoked cigarettes, richly loaded with nicotine and other compounds effective as arthropod repellents [78].

## 4. Safety: Current Pharmaceutical Drugs vs. Propolis

The arsenal of pharmaceutical drugs that have been introduced in therapy in the first half of the last century contributed largely to the improvement of human and animal health. All drugs currently used in therapy are products that underwent tests of efficacy, toxicity, preclinical, and several clinical phases until they were approved for medical uses. Nonetheless, their use is not free of the risks of adverse effects. Drugs currently used in therapy, including those of natural origin, are substances exerting high pharmacological effects. On the other hand, they have side effects that, depending on the sensitivity of the patient, may lead to fatal consequences. For example, atropine is known to cause mouth dryness, blurred vision, dry eyes, tachycardia, urinary retention, and impotency [79]. Ergotamine may lead to myocardial infarction and ischemia of limb extremities [80]. Adverse effects of quinine and chloroquine include thrombocytopenia, intravascular coagulation, hemolytic anemia, kidney injury, and cardiac ischemia [81]. Vincristine causes induced peripheral neuropathy [82]. Colchicine induces gastrointestinal disorders and multi-organ dysfunction that may lead to obit; there is no clear-cut distinction between colchicine nontoxic and lethal doses [83]. Among drugs of synthetic origin, non-steroidal anti-inflammatory drugs (NSAIDs), e.g., ibuprofen, ketoprofen, diclofenac, and piroxicam, are among the most frequently prescribed. However, they are included among the drugs with a higher risk of causing drug-induced liver injury (DILI) [84]. In many countries (Brazil included), most NSAIDs are sold as over-the-counter medicines. For this reason, despite the overall low incidence of their induced hepatotoxicity, their widescale consumption makes them relevant causes of drug-induced injury. For example, ibuprofen is regarded as an efficacious and safe NSAID, but a considerable number of idiosyncratic hepatocellular damage have been reported in the clinical literature [85]. Products derived from propolis are viewed as a possibility to treat health problems that currently are medicated with drugs causing adverse effects. Anti-inflammatory activity has been reported for several propolis types from both new and old worlds [86], including propolis types from Argentina, Chile, China, Brazil (brown, green, and red propolis types), India, Mexico, Morocco, and Portugal. If propolis were used in adjunct treatments, the doses of NSAIDs could be reduced, and the risks of liver injury decreased.

Antibiotics are relevant drugs derived from microorganisms. The introduction of antibiotics represented one of the greatest medical breakthroughs in medicine history. An outstanding corollary of the benefits brought about by antibiotics is the extension of 23 years to the average human lifespan. After the discovery of penicillin, other antibiotics derived from other fungi and bacteria have been introduced in antibacterial therapy. There are 39 clinically used antibiotics: 14 of them derive from *Actinomycetes* (e.g., kanamycin A, tetracyclines, and chloramphenicol), five from other bacteria (e.g., gramicidin A, and bacitracin A), six from fungi (e.g., amoxicillin), in addition to 14 synthetic compounds (e.g., ciprofloxacin) [87]. The search for new antibiotics continues, in part powered by the escalating resistance to antibiotics by pathogenic bacteria and fungi [88]. In the period 2017–2020, 20 antibiotics were developed, several among them already approved by the FDA [89].

Much care is advised regarding the use of antibiotics. Virtually all among them may exert side effects, sometimes with serious consequences. A long list of adverse reactions to antibiotic therapies, including hematologic effects (e.g., anemia, immune platelet dysfunction, and bleeding), drug fever, photosensitivity, hepatitis, pancreatitis, and allergic reactions (e.g., drug fever, drug rush, and anaphylactic reactions have been pointed out [90]. All chemical groups of antibiotics are prone to produce adverse effects: β-lactams (penicillin V potassium, amoxicillin, cloxacillin, cephalosporins), non β-lactams (fosfomycin, clindamycin, lisenoid, erythromycin, clarithromycin, azithromycin, methenamine, metronidazole, nitrofurantoin), quinolones (norfloxacin, ciprofloxacin, levofloxacin, moxifloxacin), tetracyclines (doxycycline, minocycline, tetracycline, vancomycin), and azoles (fluconazole, itraconazole, ketoconazole) [91]. Adverse effects reported range from allergic reactions (e.g., itching, flushing, rash, hives) to self-limiting and transient reactions (e.g., dizziness, gastrointestinal effects, nausea, candidiasis, headache, vaginitis, menstrual disorders, back pains, pharyngitis) to very serious harms (e.g., fatal anaphylaxis, renal inflammation, *Clostridium difficile* infection, hepatitis, hematuria, myelosuppression, irregular heart rhythms, pulmonary toxicity, hemolytic anemia, peripheral neuropathy). A data survey was carried out involving 1488 patients in treatment at Johns Hopkins Hospital, with a median age of 49 years. All patients were subject to antibiotic therapy for at least 24 h. It was observed that 30 days after antibiotic initiation, 298 of these patients (20%) presented at least one antibiotic-associated adverse effect. Adverse effects noticed included: nausea and vomiting, hematologic, hepatobiliary, renal, neurologic, dermatologic, cardiac, anaphylaxis, and myositis. Within 90 days of the antibiotic therapy, some patients presented clinical signs and symptoms consistent with *Clostridium difficile* infection or infection with methicillin-resistant *Staphylococcus aureus* or other problematic microorganisms [92]. This study reveals that scrutiny and objectivity are likely to demonstrate how frequently may be the emergence of adverse effects of antibiotics, especially in cases subjecting elderly patients to long treatments with antibiotics.

One of the major health problems presently faced by humanity is the growing antimicrobial resistance (AMR) by strains of pathogenic bacteria. This is far from being a new issue: in reality, as far back as 1945, Alexander Fleming warned medical authorities about the risks of the development of AMR to antibiotics. The problem has been aggravated by many inappropriate practices of antimicrobial therapies, including the prescription of antibiotics to treat viral diseases. Globalization contributes to the rapid spreading of localized episodes of AMR. Another relevant factor fostering antimicrobial resistance is the antimicrobial use in the livestock sector, responsible in the USA for 80% of the total consumption of antibiotics [93]. The burden brought about by drug-resistant infections on the public health and economy of all countries is astounding. The load is escalating with time and a review commissioned by the UK government, carried out in 2014 and 2016, forecasted that antimicrobial resistance is expected to kill 10 million people annually by 2050 [94]. A recent publication based on a broader study conveys results based on data collected worldwide in 2019, covering 204 countries and territories, as well as 471 million records from hospitals and experts about infection treatments and corresponding outcomes. Statistical analysis revealed that 4.95 million deaths associated with bacterial antimicrobial resistance were registered in 2019, among which 1.27 million corresponded to deaths attributable to bacterial AMR [95]. Considering that the accumulated number of deaths by COVID-19 in April 2022 was estimated to lie around 6 million, we have an approximate measure of the extent of the worldwide burden represented by AMR.

It is in the present context that propolis emerges as a useful aid in antimicrobial therapy. In vitro assays have shown that propolis has no cytotoxic effects on immune cells in concentrations up to 100 µg mL^−1^ [96], suggesting that the product is well tolerated and safe [97]. Results from several authors have established as 2050 mg kg^−1^ the LD_50_ of propolis in mice. Considering that flavonoids are often major components of propolis, it is predictable that propolis might have low toxicity: pinocembrin is not toxic at 1000 mg kg^−1^ in mice [98]. However, alerts have been raised that quercetin administered to mice at 100 mg kg^−1^ affects mitochondrial biogenesis [99]. Overall, however, propolis is assumed to be safe for humans. Adverse effects of propolis in people start emerging only at doses over 15 g day^−1^ [100].

The mild or null adverse effects of propolis are possibly a consequence of a selection by honeybees of plant sources containing resins. Resin sources selected by honeybees are plant materials devoid of substances toxic to insects. Worldwide, propolis contains no nitrogenous classes of secondary metabolites toxic to insects, such as alkaloids, glucosinolates, and cyanogenic compounds. Plant material containing toxins such as cardenolides, as well as plants loaded with high contents of tannins, are also avoided. On the other hand, plant exudates or young tissues (buds or leaf primordia) containing non-toxic exudates are liable to be licked or cut and chewed by the relatively delicate mouth apparatus of honeybees and are likely to become sources of resins. However, contrary to relationships between plants and bee pollinators, no coevolutionary and mutualistic links exist between bees and plant sources of propolis resins. Chemical and biological aspects involving the selectivity of honeybees of plant sources of resins have been published elsewhere [20,101].

Data about propolis safety must be viewed with caution because no information is available regarding differences in safety comparing distinct types of propolis. In fact, there have been reports about adverse effects of propolis on people sensitive to one of its components. In this regard, a case report was published about a man who developed acute renal failure caused by supplementation of Brazilian propolis during cancer treatment [102]. It has been suggested that CAPE (Figure 2, **I**), the main component of poplar propolis, may cause acute renal failure in sensitive patients. Lopez et al. (2015) detected cytotoxic effects of Brazilian red propolis against HaCAT human keratinocytes and BALBc 3T3. On the other hand, the most often reported adverse effect of propolis is the development of allergic reactions. Beekeepers have been affected by contact allergies. Proportions in the range of 0.76–4.04% of beekeepers affected with contact dermatitis have been registered in Poland and Germany [103]. Components of propolis may induce allergic reactions also in musicians and musical instrument makers [104]. Allergic contact dermatitis may also arise by ingestion of propolis extract [105]. In these cases, the sensitivity is called systemic contact dermatitis. Allergenic substances in food reach the target organ (the skin) following gastrointestinal absorption [106]. Allergic contact dermatitis has been reported in patients using hygiene products, such as shampoo containing poplar propolis [107]. It is believed that the number of cases of allergic contact dermatitis to propolis has increased in the past two decades, probably due to the use of hygiene, cosmetic and pharmaceutical preparations containing the product [108]. Several substances contained in propolis and in poplar buds have been recognized as allergens, chiefly caffeate esters, including CAPE (Figure 2, **I**) [109], while flavonoid aglycones are weak sensitizers [110]. A biotransformation of poplar propolis has been proposed, by submitting a resuspension of propolis to the activity of a cinnamoyl esterase of *Lactobacillus helveticus* [111]. They showed that the quantity of caffeate esters was reduced substantially, with no loss of antimicrobial activity. Not only caffeate esters are a matter of concern regarding allergenic effects. It has been claimed that beeswax (sometimes a major propolis component) also contains allergenic components [112]. Allergic reactions were noted in patch tests using either yellow beeswax (possibly containing propolis contaminants) or white beeswax. It is important to mention that most beeswax present in propolis is removed by alcoholic extraction. In case the presence of beeswax turns out to be a real problem in medicinal uses of propolis, a way to enhance the elimination of beeswax could be the treatment of propolis extracts at low temperatures.

A hypothesis may be put forward that allergic contact dermatitis caused by propolis is mostly restricted to the poplar type and its allergenic caffeoyl esters. All or nearly all cited literature related to the topic refers to poplar propolis type, very little having been reported about contact allergy induced by other types of propolis. Apart from a case report [108], no events linking Brazilian propolis allergic contact dermatitis have been mentioned. To our knowledge, no allergic contact dermatitis has been noticed in Japan, a country consuming preferably Brazilian green propolis. On the contrary, a randomized trial involving 80 lactating women and respective offspring demonstrated that the product neither improved nor worsened nonspecific symptoms while having no effect on atopic sensitization in infants. In fact, Brazilian propolis has been shown to exert anti-allergic properties due to inhibitory effects on the activation of mast cells and basophils. Preclinical studies demonstrated the effects of propolis extracts against allergic symptoms, such as inflammation, asthma, rhinitis, atopic dermatitis, and food allergy, most studies having been carried out with Brazilian green propolis [113]. It remains to be determined which components of propolis account for the anti-allergic activity. With the exception of one report [114], apparently no studies evaluating the anti-allergy effect of isolated artepillin C (Figure 2, **III**) have been carried out.

## 5. Antibacterial Activity

After antioxidant effects, the biological property most studied of propolis is its antibacterial activity. Up to 2019, the number of studies about the antibacterial activity of propolis, and the corresponding species of microorganisms, has been as follows: *Escherichia coli*: 120, *Staphylococcus aureus*: 116; *Salmonella* spp.: 50; *Pseudomonas aeruginosa*: 48; *Enterococcus* spp.: 30; *Yersinia enterocolitica*: 23; *Proteus mirabillis*: 22; *Klebsiella pneumonieae*: 19; *Strepcoccus mutans*: 17; *Staphylococcus epidermidis*: 16 [52]. Table 1 provides results of MIC values obtained against two bacteria species, *Escherichia coli* (Gram-negative) and *Staphylococcus aureus* (Gram-positive), of crude ethanol extracts of propolis from several regions of the world. The results reveal that most propolis types are active against Gram-positive bacteria. Propolis, in general, is inactive against Gram-negative bacteria or exhibits MIC values higher in tests against Gram-positive bacteria. Antibacterial activity of propolis varies widely, depending on the composition of the propolis, which is influenced by several factors [4,115]. The wide variability of MIC values is noted in Table 1, some propolis types having revealed very weak activity (Cameroon, Congo: MIC 10,000–20,000; Australia: MIC 2000; Iran: MIC 2500). Regarding propolis types from Brazil, the green and red types (derived from *Baccharis dracunculifolia* and *Dalbergia ecastaphyllum*, respectively) seem to have higher activity than the brown type. This latter type of Brazilian propolis is produced in the south of the country, often with uncertain composition and mixed botanical origin [20,116]. The same type of propolis, for example, Brazilian green, may exhibit widely distinct MIC values (85–5700). Comparing Brazilian green and red propolis types, the results of Table 1 suggest that the latter has higher activity than the former (MIC values 62–125 and 250–500, respectively; [54]). Some types of propolis stand out for their exceptionally low MIC values. Such are the cases of Taiwanese and Turkish (Anatolian) propolis (MIC values 10 and 8, respectively; [52]), as well as another Taiwanese propolis with MIC 10 (Chen et al., 2018) and propolis from the South African Cape region (MIC 6; Suleman et al., 2015). The results of two independent works about Taiwanese propolis, both reporting low MIC values, indicate that this type of propolis has promising possibilities in antibacterial therapy. Taiwanese propolis derives from the surface material of fruits of *Macaranga tanarius* (Euphorbiaceae) [26]. It contains interesting geranyl-flavanones, such as propolins (Figure 2, **VIII**; [66]). The activity of propolins against *S. aureus* accounts for the low MIC values of both Taiwanese propolis. While the propolis crude extract has MIC 10 µg mL^−1^, isolated propolin C has MIC 2.5 µg mL [66]. Regarding the South African propolis, its chemical profile is not particularly distinct from temperate propolis [65]. A Turkish propolis provided MIC 560 (Table 1; [64]). A wide-embracing study of the chemical composition of Turkish propolis involving 39 samples from all regions of Turkey revealed profiles distinct from European propolis, including epigallocatechin gallate (Figure 1, **XI**), myricetin, rhamnocitrin, diosmetin, and flavonoid glycosides [29].

## 6. Antifungal Activity

A review is available about antifungal activities of propolis [12]. Compared with antibacterial, works about antifungal effects have been considerably less frequent. Results reported so far involve chiefly Brazilian propolis, including the green and red types. Very little has been done in this respect concerning European propolis; basically, two studies have been carried out, one about French and another about Portuguese propolis. Studies have been conducted investigating the antifungal effects of other propolis types, including Argentinean propolis derived from *Zuccagnia punctata* (Leguminosae, Caesalpiniodieae) [117]. Most studies have involved species of *Candida*, including *C. albicans* (both sensitive and resistant to fluconazol), *C. guilliermondii*, *C. krusei*, *C. parapsilosis*, and *C. tropicalis*. Other genera of fungi studied have been *Aspergillus*, *Microsporon*, *Trichophyton*, and *Trichosporon*. One in vivo preclinical study was carried out about effects against a vulvovaginal standard and fluconazol-resistant isolates of *C. albicans* [118]. The main propolis components responsible for antifungal activity have been suggested to be flavonoids. It has been assumed that chalcones present in extracts of *Z. punctata* are chief antifungal components [117]. The mechanism of antifungal action is believed to occur by apoptosis through metacaspase and Ras signaling, in addition to inhibition of the expression of several fungal genes involved in pathogenesis, cell adhesion, biofilm formation, and filamentous growth [12]. Pinocembrin, a major component of propolis from temperate regions, reduces the levels of phosphorylated adenosine nucleotides of hyphae of *Penicillium italicum* and damages its cell membrane, causing ionic leakage and loss of soluble proteins [119].

## 7. Antiviral Activity

It is urgent the introduction of new antiviral agents in therapy due to several reasons: (1) the damage to the human population worldwide caused by virus infections, such as COVID-19, AIDS, Ebola, influenza, dengue, and chikungunya; (2) the paucity of available antiviral drugs; (3) the speed of development of genetic changes and resistance by the viruses. Viruses are obligatory intracellular pathogens. Antiviral effects of propolis have been studied, promoting cell infections in vitro. The activity is evaluated using different concentrations of propolis to treat the infection. The results obtained may be evaluated by different methods: polymerase chain reaction (PCR), real-time (RT)-PCR, cytotoxic assays, cytopathic effects, and neuraminidase inhibition, among other procedures. Evaluation of the antiviral activity of propolis started in the 1960s, and the number of studies has increased since then. A review was published involving several viruses and the activity of propolis from different regions of the world [98]. The most studied virus has been herpes simplex, then influenza. Other viruses have been less studied. Among substances that may be found in propolis, rutin has the highest binding energy to ACE2, followed by myricetin, CAPE (Figure 2, **I**), hesperetin, and pinocembrin. PAK1 is a kinase with an important role in viral infections. Artepillin C (Figure 2, **III**), a major component of Brazilian green propolis, selectively inhibits PAK1. Table 2 contains a list of reports about the antiviral activity of propolis.

From data in Table 2, it may be concluded that propolis protects the organism against viral infection by several mechanisms, including inhibition of absorption by host cells and inhibition of viral replication, damage to the viral structure, and activation of synthesis of cytokines. The results obtained so far are encouraging, and much remains to be done. Recently, because of the COVID-19 pandemic, SARS-CoV-2 has been the aim of studies to evaluate the efficacy of propolis against the virus. Reports have been published about anti-SARS-CoV-2 activity based on inhibition of the angiotensin-converting enzyme 2 (ACE2), to which SARS-CoV-2 strongly binds for invasion and replication inside host human cells [136,137]. The binding energy to ACE2 of several flavonoids and other propolis components has been evaluated [138]. Rutin has the highest binding energy, followed by myricetin, CAPE (Figure 2, **I**), hesperetin, and pinocembrin. Catechin and *p*-coumaric acids are also efficient at inhibiting ACE2 [139]. Molecular docking studies (a type of in silico research) indicate that rutin and CAPE have the highest affinity with ACE2 among common propolis components [140]. Another study based on molecular docking suggests that sulabiroin-A (Figure 2, **XII**) may be an inhibitor of SARS-CoV-2 [141]. Sulabiroin-A is a podophyllotoxin derivative recently isolated from propolis of the stingless bee *Tetragonula* aff. *biroi*, from South Sulawesi, Indonesia [142].

## 8. Synergy with Other Antimicrobial Substances

Propolis has many secondary metabolites with a broad spectrum against a wide diversity of microorganisms. It exhibits strong and multi-targeted activity attributed to many of its phenolic components [143]. It has been shown that propolis has the capacity to enhance the efficacy of other antibacterial agents, such as honey and antibiotics [11]. Mixing propolis with another antibacterial substance often results in synergy: the outcome of the combination is more powerful than the effects of any of the products alone. An additional advantage of the approach is its multi-target characteristic due to distinct mechanisms of action of the propolis constituents and of the other antimicrobial product. Thus, the synergy between propolis and other products has opened the possibility of attaining control of resistant strains of pathogenic bacteria, such as methicillin-resistant varieties of *S. aureus*. Several products known to have antibacterial properties, such as medicinal plants, honey, and antibiotics, have been used to interact with propolis in tests against pathogenic microorganisms, and positive effects have often been obtained.

### 8.1. Propolis and Honey

Honey has long been used to treat wounds, including chronic ulcers. There have been reports of clinical use of honey dressings to treat ulcers over the last decades; more recently, the number of uses increased substantially [144]. Honey has a broad spectrum of antibacterial activity and is active against pathogenic microorganisms affecting chronic wounds, such as *Pseudomonas aeruginosa* and *S. aureus* [145,146]. Manuka honey, produced in Australia and New Zealand with nectar from flowers of *Leptospermum scoparium* (Myrtaceae), has been the most extensively studied type of honey regarding composition and medicinal properties. The antibacterial effect of honey derives from some of its physicochemical characteristics, such as high sugar concentration, low water activity, and low pH, in addition to the presence of hydrogen peroxide [146]. Manuka honey, however, is unique for possessing methylglyoxal and leptosin (a glycoside). These honey components are responsible for the non-peroxide antibacterial activity of the product [147,148,149]. Manuka honey exhibited an expressive MIC value (8 µg mL^−1^) against strains of *S. aureus* [146]. However, MIC values 3.1–6.3 have been obtained for honey derived from the Chilean Ulmo tree (*Eucriphia cordifolia*, Cunoniaceae) [150]. MIC values 9.38–18.75 were obtained regarding honey from Ethiopia against *S. aureus* [151]. Table 3 lists works involving propolis and honey from different countries and their effects on several microorganisms.

### 8.2. Propolis and Antibiotics

Many approaches over the last decades have been conducted to test the capacity of propolis to interact with antibiotics, aiming to break the resistance of pathogenic microorganisms and reduce the doses of the drugs. Propolis possesses a strong and multi-targeted antibacterial activity, chiefly attributed to its flavonoids. A study with positive results evaluated the synergy between antibiotics and a polyphenol-rich mix composite with plants and propolis extracts [155].

The first widely cited study about synergism between propolis and antibiotics involved European propolis and several antibiotics, working in synergy against *S. aureus* [156]. An extensive list has been published, involving tests with a diversity of antibiotics, microorganisms, and propolis from distinct localities [157]. The authors commented that most studies have enhanced the effects promoted by propolis extracts by comparing zones of inhibition or MIC values rather than determining the FIC index, a parameter that indicates synergy if the result is ≤0.5. Although a few of the works in the mentioned failed to reveal the interaction between propolis and antibiotics, in most cases positive results of synergism between propolis and antibiotics have been obtained. The list includes results involving a wide diversity of antibiotics against species of Gram-positive and Gram-negative bacteria. The most studied propolis type in the list is Brazilian green, followed by Brazilian red propolis. Several studies have used propolis from Poland and other European countries. Apparently, no studies of synergy have been conducted with propolis from South America other than Brazilian propolis. No studies have included propolis from Africa, Australia/New Zealand, and Pacific propolis (Okinawan and Taiwanese propolis). Given the known high activity of the latter type of propolis and of its components [67], studies evaluating the synergy of this type of propolis with antibiotics are expected to render promising results.

Not only in vitro studies combining the effects of propolis and antibiotics have been performed, but also tests observing results of treatment of mice infected with *Salmonella enterica* (a Gram-negative bacterium), with reduction in bacterial load and improvement in the survival rate, hematological parameters, and conditions of the kidney, spleen, and liver of the animals [158]. Among the bacteria that have been used in the tests, predominate *S. aureus*, both sensitive and methicillin-resistant strains. Several Gram-negative bacteria are included in the list of the studies, such as *E. coli*, *Haemophilus influenzae*, *Helicobacter pylori*, and *Salmonella thyphimurium*, often with positive results.

### 8.3. Propolis and Antifungal Drugs

The most frequently studied species of pathogenic fungi are the yeasts of the genus *Candida*, followed by *Cryptocarius neoformans* and the mold *Aspergillus fumigatus*. Pathogenic fungi cause superficial mucosal infections, such as oropharyngeal and vaginal candidiasis, as well as life-threatening systemic infections, for example, disseminated candidiasis, cryptococcal meningitis, and invasive aspergilosis [159]. Available classes of antifungal drugs have distinct mechanisms of action. Amphotericin promotes the alteration of membrane function; flucytosine inhibits DNA and RNA synthesis; azoles (fluconazole and itraconazole) and nystatin inhibit ergosterol biosynthesis; echinocandins inhibit glucan synthesis [160]. Similar with the bacterial resistance, the resistance of fungi to the available antifungal agents is a serious concern worldwide; it includes novel variants previously susceptible (e.g., the mold *Aspergillus fumigatus*) and several species that are increasingly acquiring resistance to multiple antifungal drugs, as is the case of *Candida auris* [161].

Little has been done regarding the synergism of propolis and antifungal drugs. Positive results of synergism between 13 samples of Serbian propolis and nystatin against *Candida albicans* has been obtained [162]. Pippi et al. (2015) It has been observed that *n*-hexane extract of Brazilian red propolis inhibits the growth of fluconazole-resistant strains of *Candida glabrata* and *C. tropicalis* [163]. They obtained synergistic effects between the propolis extract and fluconazole against both species but no synergism between propolis and anidulafungin. The authors suggest that the absence of synergism with this drug may be due to a similar mode of action on the fungal membrane by both red propolis and the antifungal drug.

### 8.4. Propolis and Antiviral Drugs

Despite the emergence of recent outbreaks of severe viral diseases, such as dengue, chikungunya, and AIDS, in addition to the severity of the worldwide COVID-19 pandemic, very little has been done to evaluate synergistic effects between propolis and current pharmaceutical antiviral drugs. Apparently, no studies about synergistic effects have been done between propolis and remdesivir. The composition of the propolis from the Hatay region (south of Turkey) has been chemically characterized [164]. Typical components of poplar propolis were detected, such as CAPE and galangin (Figure 2, **I** and **II**, respectively), as well as chrysin. The authors evaluated the effects of the hydroalcoholic extract of the propolis on the replication of herpes simplex virus types 1 and 2 and verified that the effect was similar with acyclovir. Further tests combining propolis and acyclovir resulted in enhancement of the inhibitory effect.

## 9. Gaps in Studies about Antimicrobial Activity of Propolis

The bibliography about the antimicrobial activity of propolis is abundant. However, bearing in mind the amplitude of the diversity of both propolis types and pathogenic microbes, much remains to be done toward the evaluation of the antibacterial, antifungal, and antiviral activity of propolis types so far uninvestigated in this regard. Several types of propolis from active apicultural centers, such as Brazil, Australia, and New Zealand, have been poorly studied regarding their antimicrobial potential. Despite interesting results obtained testing the efficacy and antimicrobial activity of manuka honey [165], little or nothing has been done about the antimicrobial activity of propolis types from Australia and New Zealand. It has been commented on the high activity of geranylated flavanones, such as propolin A, from *Macaranga tanarius* propolis. Synergistic effects of this type of propolis with commercial antibiotics, antifungals, and antivirals may provide interesting results. A new type of green propolis from northeast Brazil has been reported, containing a wide diversity of flavonoids, including a high content of chalcones [44]. No studies about the antimicrobial activity of this promising type of propolis have been done so far. Other propolis types containing chalcones are produced in Argentina, derived from *Zuccagnia punctata* (Leguminosae, Caesalpinioideae) [43]. Promising antibacterial [51] and antifungal [117,166] activities have been observed. Apparently, neither studies of antiviral activity nor of synergy with antibiotics and commercial antifungal drugs have been performed with this type of Argentinian propolis. Several papers have been reported about African propolis, but only a few have been evaluated regarding their antimicrobial activity. As seen above, very little has been done about the synergy of propolis and commercial antifungal and antiviral drugs.

## 10. Propolis Uses in Therapy against Infectious Diseases: Clinical Trials

### 10.1. Standardized Propolis Extracts

Prospects of propolis eventually turning out to be a prolific source of useful new drugs in modern medicine are common in propolis literature [4]. The intense investigation of the biological properties of isolated constituents of several propolis types is in line with this perspective. However, because of the wide variability of the composition of propolis (even among propolis of the same type), the precise composition of an extract of a determined sample of propolis is unpredictable. This is a serious constraint for using propolis in modern clinical dentistry and medicine. Clinical treatment of patients is normally performed using pharmaceutical products with definite composition. Such accuracy is impossible in medicines prepared with propolis products. Aware of these limitations and aiming to introduce propolis in clinical therapy, propolis scientists and entrepreneurs have developed procedures to obtain standardized propolis extracts. Among them, some have been used in clinical trials, and several are commercially available. Among the objectives of all procedures, one of them is to eliminate biologically inert propolis components, such as wax and pollen. EPP-AF^®^ is a standardized propolis extracted obtained from Brazilian green propolis [167]. A topical delivery system, PE-8, containing extract of Brazilian green propolis, was developed for the treatment of herpetic lesions. The product contains 8% of crude isopropanol/ethanol extract of Brazilian green propolis, incorporated in a mixture of P407 (a surfactant) and C934P (an acrylic acid polymer), with pH adjusted with triethanolamine [123]. The standardization of the product was based on the content of total phenolic content. GH 2002 is a viscous extract of propolis from the Czech Republic, obtained after a procedure to remove wax and resinous material, containing about 50% of the initial raw propolis [59]. Propoelix^TM^ is a product obtained by superextraction of poplar propolis, using a procedure that removes inert components (wax, resinous material), leaving behind propolis active ingredients in a water-soluble form [168]. M.E.D.^®^ is a standardized extract also derived from poplar propolis, using a patented process named “multi-dynamic extraction”, by which a reproducible chemical composition is obtained, specially concerning the flavonoids galangin, chryxin, pinocembrin, apigenin, pinobanksin, and quercetin [169]. It has been claimed that the procedure enables attaining both reproducible chemical composition and antimicrobial activity, independently from the chemical composition of the starting propolis [170]. A procedure has been described to produce sHEP, a hydroalcoholic extract obtained from poplar Eurasian propolis, by extraction with ethanol 80% and standardized by determination of the galangin content [130,136]. The use of propolis liposomes (PP-Lip) has been suggested for the improvement in cellular uptake by endocytosis. PP-Lip was prepared by dissolving lipoid S75 (70% phosphatidylcholine mixed with fat-free soybean phospholipids), cholesterol, and propolis extract in the least volume of absolute ethanol. The solution of lipids was sprayed on the surface of an aqueous solution of 9% sucrose. After evaporation of excess ethanol, liposomes were formed spontaneously [171]. Ear drops were developed for pharmaceutical use, containing extract of propolis derived from *Zuccagnia punctata*. Dry extract of propolis was included at the concentration of 2% (*w*/*v*) into two viscous solvents (ethanol:propylene glycol and ethanol:glycerol:propylene glycol) [52].

Propolis phenolic composition is complex. The synergistic way of action among the components of propolis [37] motivates assumptions that different propolis fractions may be isolated from a propolis sample, each of which with distinct phenolic composition and distinct biological activity. Thus, an alternative way, although probably costly, to obtain mixtures of propolis components with highly bioactive substances and precise composition may be the fractionation of propolis extracts. In fact, several procedures for propolis fractionation have been proposed. Supercritical antisolvent fractionation of tinctures of propolis from New Zealand was used to obtain a flavonoid fraction as the primary product and an essential oil/ethanol fraction as a secondary product [172]. Propolis fractions may be obtained by nanofiltration [173]. Fractions with different compositions were obtained, with distinct contents of flavonoids and antioxidant activity ranging from 19% to 98%. Fractionation of Slovenian propolis in solid-phase extraction has been performed, using a polymeric reversed-phase cartridge and ethanol solutions with increasing concentrations [174]. Five eluates were obtained, with distinct total phenol content and antioxidant activity. Fractionation of Brazilian red propolis was carried out using supercritical extraction [175]. 

Much remains to be done, aiming to explore in detail the possibilities of obtaining mixtures of propolis components with potent medicinal properties. The holy grail expected from this enterprise is a highly potent blend of propolis components, containing a limited number of constituents and precisely known composition. Once known the most convenient propolis fraction, the production of a combination of active ingredients could be enhanced and the costs reduced by obtaining the active substances from rich natural sources or synthesis. For example, galangin (Figure 2, **II**) is abundant in rhizomes of *Alpinia officinarum* (galangal, an Asian traditional herb) [176]. Total synthesis on laboratory scale has been achieved for CAPE (Figure 2, **I**) [177,178], as well as for artepillin C (Figure 2, **III**) [179,180,181].

As commented above, one of the main objectives of the production of standardized propolis extracts or fractions of propolis active constituents is to eliminate inconvenient propolis components. In this regard, it is contradictory to the context involving an important poplar propolis component: CAPE (Figure 2, **I**). This caffeoyl ester is one of the most studied propolis components, with a multitude of biological effects, many of which with medicinal possibilities [182,183]. On the other hand, it is prudent to consider its role in inducing contact allergic dermatitis [109]. In case advances are taken toward the production of propolis mixtures derived from poplar propolis applicable in clinical therapy, the alternative is to produce distinct mixtures, some containing CAPE (applicable to patients not sensitive to the substance) and others devoid of the substance (for patients allergic to it).

### 10.2. Dentistry Clinical Therapy

Reviews are available about the usefulness of propolis for the prevention and treatment of caries, gingivitis, periodontitis, and other health problems related to dentistry [12,184]. Clinical trials have shown that propolis present in mouthwash and toothpaste significantly reduces plaque in the gingival baseline. Studies based on patients with cleft lip and palate treated with fixed orthodontic appliances provided positive results of decrease in parameters of OPI and GI and percentages of pathogenic oral bacteria, with improvement in oral health, resulting from antibacterial, anti-inflammatory, and regenerative properties of propolis. In a randomized clinical trial, propolis has shown efficacy in improving the periodontal status and glycemic control in patients with type 2 diabetes mellitus and chronic periodontitis. Several clinical experiments have shown the efficacy of propolis in endodontics and the treatment of periodontitis.

### 10.3. COVID-19

Medicinal clinical trials using propolis have not been as frequent as in dentistry. The COVID-19 pandemic encouraged some medical teams to test propolis or honey as adjunct treatments. A detailed review has been published about the prospects of honey and propolis for the treatment of COVID-19 based on in vitro, in silico and clinical studies [138]. The study focuses on known properties of phenolic components of propolis, including CAPE (Figure 2, **I**), artepillin C (Figure 2, **III**), rutin, naringin, and luteolin. Such components may inhibit viral spike fusion in host cells, establish viral-host interactions and trigger the cytokine storm and viral replication. Evidence has been raised about the interactions of these propolis components with SARS-CoV-2 proteins, with possible antiviral effects. The review mentions clinical treatments of COVID-19 patients with the adjunct use of honey combined with herbal products.

A case report gives account of a patient with a positive test for COVID-19 and a 2-day history of several symptoms of the disease. Upon treatment with EPP-AF^®^ over 12 days, the patient recovered from the COVID symptoms, and a test by RT-PCR gave a negative result [185]. Despite the encouraging results obtained, the authors of the report admit that no reliable conclusion may be drawn about the involvement of propolis in the treatment of COVID-19 from a single case report.

A randomized, open-label, single-center, controlled clinical trial enrolling 124 hospitalized patients in the treatment of COVID-19 was conducted, using EPP-AF^®^ as an adjunct therapy [186]. Doses used were 400 and 800 mg day^−1^ for groups 1 and 2 (40 and 42 patients, respectively). The control group comprised 42 patients. Benefits derived from the use of propolis were noted regarding shorter length periods of hospital stay post intervention (7 days for group 1 and 6 days for group 2, compared with 12 days for control) and a lower rate of acute kidney injury in groups treated with propolis. No adverse events were observed derived from propolis use. The authors were not able to specify the mechanisms involved in the benefits provided by propolis in their trial. They suggest that the known anti-inflammatory and immunomodulatory effects of propolis were helpful concerning the improvements of the lung and kidney conditions of the patients.

The results obtained so far are encouraging toward the clinical usefulness of propolis and hopefully will stimulate other professionals to test propolis in dentistry and medical treatments.

## 11. Propolis Production and Beekeeping: Environmental and Social-Economic Benefits

Among apiary products, propolis stands out in several parts of the world for the recent increases in demand and production. A subject of much concern nowadays is the gradual decrease in the number of bees in America and Europe. This is one of the many environmental crises that threaten biodiversity in the present day. A document published by FAO [187] alerts that the current decline of bee populations in several parts of the world represents a threat to the production of nutritious plant foods, such as fruits, nuts, and many vegetables. In case the current fall in the number of bee population prevails, these healthy sources of functional chemicals tend to be substituted by staple crops such as rice, corn, and potatoes. The declining number of populations of honey and stingless bees has been attributed to intensive farming practices, mono-cropping, excessive use of crop defenses, and climate changes. Beekeeping contributes to crop pollination and increases the quantity and quality of fruits, nuts, and oils [188]. In general, apiculture is regarded as a self-sustainable and environmentally friendly activity of production of animal-derived goods [189], quite distinct from cattle ranching, for example. However, concerns have been raised about the negative ecological effects of concentrating focus only on introduced insect species (*Apis mellifera*). The outcome may have harmful consequences for populations of native bee pollinators [190,191]. Thus, beekeeping based on stingless bees is viewed as necessary to counteract the aggressive competition of honeybees for nectar and pollen. Fortunately, meliponiculture, including the production of honey and propolis of stingless native bees, is a growing market in several parts of the tropical and subtropical world, including Brazil and Australia [192].

Beekeeping in Brazil and other developing countries is practiced not only by large companies but also by small family apiaries. It requires a low initial investment, stimulating the fixation of people in the field. Contrary to other activities involving livestock, beekeeping requires no daily care and thus allows small rural farmers to combine the production of goods from the apiary with other farming activities, thus attaining an alternative and additional income source [176,189]. An outstanding example of the environmental and social-economic benefits derived from beekeeping in Brazil is the introduction and posterior spread of the production of red propolis in the littoral of the state of Alagoas (northeast Brazil). A considerable number of producers of red propolis in the locality were formerly crab fishers. The shift to beekeeping by these people represented a notable gain in their income and living conditions. The increase in the production of red propolis represented as well an environmental gain, motivated by a reduction in predatory crab fishing and improvements in the protection of the areas of mangroves in the Brazilian northeast. Red propolis is the second most important type of Brazilian propolis, being exported in increasing amounts to several countries, chiefly China. There has been a spread of production of red propolis in mangroves of several states of northeast Brazil, aggregating the labor of in-need people, including local indigenous families, with technical support by official agencies and enterprises.

## 12. Conclusions

Several decades of scientific investigation by researchers from many parts of the world have gathered a plenitude of data indicating that propolis may be helpful and safe for mankind in a similar way it has been useful over millions of years for bees: a potent ally in the struggle against microbial pathogens. The reliability of the evidence compiled over these years, allied to the concerns brought about by microbial resistance, the speed with which pandemics have spread from country to country and from continent to continent, as well as the severity of some infectious diseases, has convinced many scientists and professionals from the health area about the convenience of considering the introduction of propolis products as medicines or adjuncts in clinical treatments. Although still incipient, the results that have emerged are encouraging, permitting prospects that propolis is about to be promoted to a position in medicine of a player not only in traditional, complementary, and alternative medicine but also as a relevant protagonist in clinical treatments. The culmination of this achievement requires further progress toward the development of products based on active propolis components with desirable characteristics of water solubility and definite composition. Most known potentialities of propolis in therapy against infectious diseases have been drawn from studies dealing with Brazilian green and temperate propolis types. Much remains to be done regarding other known types of propolis, particularly several types containing substances with high activity against microorganisms. Clinical tests of adjunct therapy with antibiotics and antivirals still need to be carried out with several promising types of propolis. Standardized extracts of propolis have shown their utility in clinical tests. Promising possibilities are expected from approaches of fractionation of propolis, aiming to obtain mixtures of active components with precisely known composition, high antimicrobial activity, and no adverse effects.

## Figures and Tables

**Figure 1 molecules-27-04594-f001:**
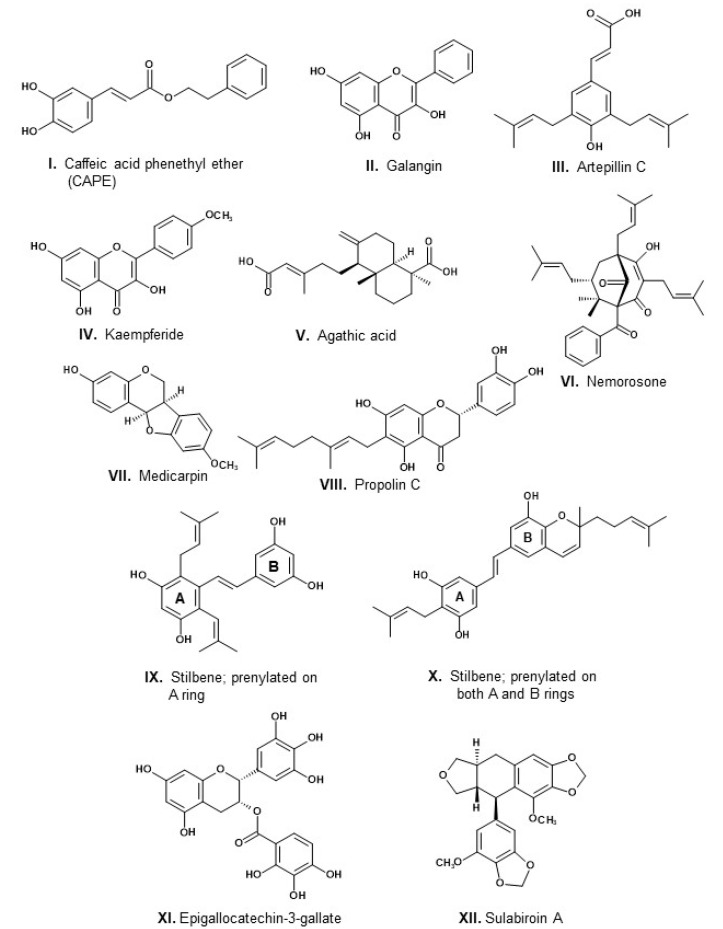
Structures of components of propolis. For correspondence between components and propolis types, see text.

**Figure 2 molecules-27-04594-f002:**
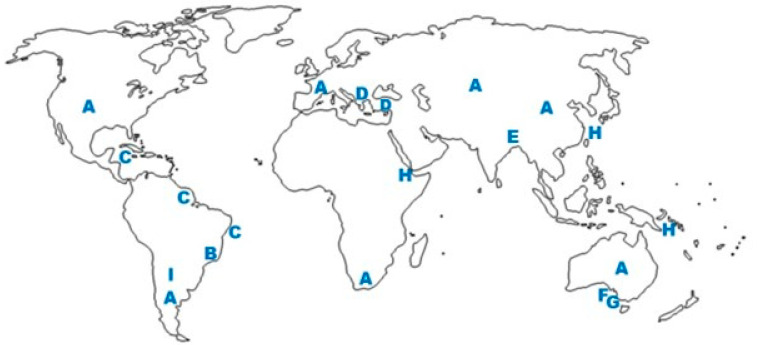
Examples of propolis types with known botanical origin and corresponding world location. **A**: temperate propolis, derived from *Populus* species; **B**: green propolis from southeast Brazil, derived from *Baccharis dracunculifolia*; **C**: red propolis from the littoral of northeast Brazil, derived from *Dalbergia ecastaphyllum*; **D**: Mediterranean propolis, derived from conifers and other plant species; **E**: Nepalese propolis, derived from *Dalbergia* sp.; **F**, **G**: propolis types from Kangaroo Islands, derived from *Leptosperma* sp. and *Acacia paradoxa* [27,31]; **H**: propolis Okinawa, Taiwan, Kenya [32] and Solomon Islands [33], derived from *Macaranga* spp.; **I**: Argentinean propolis, derived from *Zuccagnia punctata*.

**Table 1 molecules-27-04594-t001:** Values of minimal inhibitory concentration (MIC_50_, µg mL^−1^) of ethanol extracts, against two species of bacteria, of propolis from several regions of the world.

	Bacteria Species		
Propolis Origin	*Escherichia coli* *	*Staphylococcus aureus*	References
Africa (Cameroon, Congo)	Inactive	10,000–20,000	[49]
Argentina	-	50–800	[50,51]
Australia	-	1200	[52]
Australia	-	900	[53]
Australia	-	400	[54]
Australia	-	2000	[54]
Brazil	-	612	[52]
Brazilian brown	Inactive	inactive	[55]
Brazilian green	-	85–5700	[56]
Brazilian green	Inactive	250–500	[55]
Brazilian red	12.5–200	24–100	[57]
Brazilian red	-	280	[39]
Brazilian red		125	[25]
Brazilian red	Inactive	62–125	[55]
Bulgaria	-	125	[52]
Chile	-	1,445	[52]
Cuba	Inactive	4.4–58.2	[58]
Czech Republic	-	600	[52]
Czech Republic	-	130–500	[59]
Czech Republic	-	600–1200	[48]
Germany	-	750	[52]
Germany	-	1200	[48]
Greece	-	393	[52]
Hungary	-	100–400	[60]
India	-	500	[52]
Iran	-	2500	[61]
Ireland	-	545	[52]
Ireland	-	80–600	[48]
Italy	-	620–2500	[62]
Morocco	-	360	[52]
Oman	-	81	[52]
Poland	-	555	[52]
Poland	-	390–780	[63]
Russia	-	256	[64]
Slovak Republic	-	255	[52]
South Africa, Cape	781	49	[65]
South Africa, Cape	781	6	[65]
South Africa, Natal	781	195	[65]
South Africa, Pretoria	781	24	[65]
Taiwan	-	10	[52]
Taiwan	Inactive	10	[66]
Turkey	-	8	[52]
Turkey	-	560	[64]

*—: unavailable data.

**Table 2 molecules-27-04594-t002:** Reports about propolis antiviral activity.

Virus *	Propolis Type	Dose	Mechanism	Reference
HSV-1	Poplar	72 µg mL^−1^	Inhibitory of synthesis of virus DNA	[120]
HSV-1	-	LC_50_: 0.5% propolis extract	Inhibition of absorption by VERO cells and replication	[121]
HSV-1	Brazilian green	10 mg kg^−1^	Reduction in virus titer in infected mice	[122]
HSV-1	Brazilian green	-	Damage to virus structure	[123]
HSV-2	Czech	4 µg mL^−1^	Damage to virus envelope	[124]
HSV-2	Czech	4 µg mL^−1^	Inhibition of absorption by RC-37 cells	[125]
HSV1/2	Poplar	100 µg mL^−1^	Inhibition of absorption by RC-37 cells	[126]
HSV1/2	Poplar	25–200 µg mL^−1^	Inhibition of viral replication	[127]
HSV-2	Brazilian brown	50 mg kg^−1^ in mice	Anti-inflammatory and antioxidant effects	[128]
H3N2	Poplar	50 µg mL^−1^	Inhibition of viral replication	[129]
H1N1	Bulgarian	100 µg mL^−1^	Inhibition of viral replication	[12]
H7N7	Bulgarian, Brazilian, Egyptian, Mongolian	4–35 µg kg^−1^ in DBA/2 mice	Increase in IFN-γ and Th1 activation	[34]
H1N1	Poplar	35 µg mL^−1^	Stimulus of secretion of IL-6 and IL-1β	[130]
HTLV-1	Poplar and CAPE	-	Prevention of *TAX* oncogene and activation of NF-κB	[131]
PV-1	Brazilian green	-	Damage to viral cycle	[132]
HIV	Brazilian	-	Inhibition of reverse transcriptase	[10]
HIV	Brazilian	-	-	[133]
HIV	Poplar	-	Replication suppression	[134]
SARS-CoV-2	Poplar	12.5–25 µg mL^−1^	Inhibition of replication	[135]

* HSV: herpes simplex virus; H_x_N_y_: influenza viruses; HTLV-1: human T-cell leukemia-lymphoma virus; PV-1: poliovirus (responsible for poliomyelitis in humans).

**Table 3 molecules-27-04594-t003:** Examples of tests with positive results of synergy between extracts of propolis and honey.

Microorganism	Locality	Reference
*Bacillus megaterium*	Portugal	[152]
*B. subtilis*	Portugal	[152]
*B. cereus*	Portugal	[152]
MSSA	Portugal	[152]
MRSA	Portugal	[152]
MRSA	Egypt	[153]
*Escherichia coli*	Saudi Arabia	[154]
*Staphylococcus aureus*	Saudi Arabia	[154]
*Candida albicans*	Saudi Arabia	[154]
*E. coli*	Egypt	[154]
*S. aureus*	Egypt	[154]
*C. albicans*	Egypt	[154]

## Data Availability

Not applicable.

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
