# Peer review of "Perspectives for Uses of Propolis in Therapy against Infectious Diseases"

_molecules, 2022, doi:10.3390/molecules27144594_

Round 1

Reviewer 1 Report

General comments-

The current review article targets a unique topic with wider pharmaceutical potentials. Overall, the manuscript is well written with supporting literature reviews and summarized data. Still, there are a few sections as highlighted in the comments needed to be addressed, therefore, I recommend the manuscript can be accepted for the publication with “minor revision”.

Comment 1: Table 1, Author may briefly explain the difference between Brazilian green, Brazilian red and Brazilian brown type of propolis in the respective text for better clarity to readers.

Comment 2: Lines 194-262, The background about risk and harmful effects of current pharmaceutical drugs it too descriptive. Author needs to rewrite this section critically highlighting the drawbacks of major pharmaceutical drugs and how can propolis be used to overcome such limitations. It is also essential to provide possible solutions to minimize the side effects and other allergic responses of propolis consumption. 

Comment 3: Lines 276-279, It is interesting to notice that honeybees have ability to avoid nitrogenous metabolites and other toxic plant metabolites while selecting the source of plant resin. It would be interesting to see the biological and/or molecular insight of this interaction. Author may update the information about research carried out in this direction.  

Comment 4: Lines 741-755, In the conclusion, it important to highlight the major research gapes and challenges associated with propolis needed to be addressed. Additionally, author should provide some future recommendation for fellow researchers which is lacking in the conclusion of current manuscript.  

Comment 5: In general, I think a diagram or a table describing the major types of propolis and their main characteristic will be very useful for wider community of readers.

Some minor corrections/ comments

Comment 1: Line 29, error in inverted comma.

Author Response

Modifications introduced in the revised version are outlined in yellow, including new references. Modifications introduced, but not originated from comments of the reviewer, are outlined in grey.

The author thank the overall positive appreciation by the Reviewer and the constructive comments and suggestions.

Comment 1: the differences between the three types of Brazilian propolis were briefly outlined in the revised version between lines 396-401.

Comment 2: additional text is presented in the new version, touching this point. Two additional texts are seen between lines 229-243 and 272-283.

Comment 3: this point was treated in detail in the reference Salatino and Salatino (2021). Brief new information was added in the revised version, lines 323-328, with the comment that biological and chemical aspects in the relatioship between plants and bees are detailed in the reference.

Comment 4: section “Conclusion” was extended to indicate gaps in topics related with the paper and recommendations dealing with points deserving attention (lines 805-814).

Comment 5: a figure was included (Figure 1), presenting a world map, with indication of localities of production of propolis types, for which corresponding plant sources of resin have been determined.

Minor comment: error about an inverted comma on the first line of Introduction was corrected.

Please see attachment containing the revised version.

Reviewer 2 Report

Perspectives for Uses of Propolis in Therapy Against Infectious Diseases

General comment

The review paper:” Perspectives for Uses of Propolis in Therapy Against Infectious Diseases” of  Antonio Salatino, represents valuable contribution toward comprehensive overview of propolis characterization and its uses and potential for further uses in human and domestic animal therapy against infectious diseases. The review is well structured and very well covers literature sources. It addresses the topic of propolis use as an anti-infectious agent in  synthetic and associative manner, pointing out venues that still prevents full inclusion of propolis as formal therapeutic agent of mainstream medicine.

However, before being suitable for acceptance and publication, author should clearly point out advantages and novelty of this current review compared to the recent literature overview falling in the same topic.

In addition, please find the following major and minor comments so to improve clarity, understanding, associative links and missing references.

Major comment

As point out in the general comment already, in the introduction at its very end, a consize paragraph should be inserted, outlining what novelty this review within a given topic is providing to its readers. What is that what is unique and beneficial for readers that other reviews such as

1)      Propolis as a novel antibacterial agent by Mohammed Saad Almuhayawi in Saudi Journal of Biological Sciences 2020 https://doi.org/10.1016/j.sjbs.2020.09.016 or

2)      The use of propolis in dentistry, oral health, and medicine: A review.  DOI: 10.1016/j.job.2021.01.001

For example, a review mentioned here under the 1) was cited by author outlining only “It has been shown that propolis has the capacity of enhancing the efficacy of other anti-bacterial agents, such as honey and antibiotics (Almuhayawi 2020)” which is loose and relative since it is not related to specific original articles underlying this. A more precision in conveying the messages from the studies should be undertaken and to acknowledge that this is also a review paper…

Minor comments

1)       Abstract – last sentence should be rewritten so to avoid the term “unprivileged”. Also if there is such last sentence in the abstract pointing Brazil, this should be fortified within the main text with appropriate discussion.

2)       There are no ends with “” signs in the first sentence of the Introduction. Let author take good care of manuscript’s typos.

3)       For the sake of broader life science and medical community readers, it would be nice to explain somewhere in the beginning (introduction), that there are several types of propolis according to the color, like green and red DOI: 10.1007/s11356-019-07458-z  and not to be mixed with other definitions, such as orange and blue propolises mentioned in https://doi.org/10.1016/j.lwt.2018.04.063

4)       Figure caption. At present it states:” Figure 1. Structures of components of propolis. For correspondence between components and propolis types, see text.”

It might be more clear to write like this:

” Structures of the major phenolic components present in various propolis types listed in the text.”

5)       Please take care of numbering on propolis types in “Propolis Composition” part. There are two time “e” and therefore it is confusing. Consider even changing your figure to secure grasping the relationship between major propolis types and their major molecular constituents!

6)       Lines 166-168 – Would disagree with the link that major reason for propolis not being recognized as formal medicine and “ substantially produced” by pharma sector caused by uncertainty in its beneficial medical effects would be “repaired” by understanding apies biology. I meant that it is of course desirable to fully understand nature and consequently biomimicry that can then be better explored and exploited. I just disagree that this will remediate propolis utilization by mainstream pharma/medical sector. To convince them, it is only possible through controlled clinical studies with human or domestic animal subjects, that luckily author dedicated a full chapter. I guess this can easily be corrected with change of sentence flow in this section. Again, it would be good that a native English-speaking person exposed to life science edits the text.

7)       Safety: current pharmaceutical drugs x propolis – I do not understand “x” in this title. Please make clear and easily understandable title

8)       If it is possible to make Conclusion more concrete, based on the all previous chapters. I cannot see any take away message in the conclusion from the last paragraph concerning “Propolis production and beekeeping: environmental and social-economic benefits”.

Wish you luck in your professional work

Kind regards

Author Response

The author thank the overall positive appreciation of the paper by the Reviewer and the constructive comments and suggestions.

Following the recommendation of the Reviewer, the last paragraph of the Introduction was extended to explain the originality of the paper, in comparison with previously published reviews dealing with similar topics. The two references indicated by the Reviewer are cited in the submitted review. The text ammended is outlined in blue in the new version.

Minor modifications were introduced, not originated by comments of the Reviewer. They are outlined in grey.

Please see the attachment, containing the revised version.

Reviewer 3 Report

Comments and suggestions for Authors 

The collected literature data on propolis in the reviewed manuscript can be used in practice, both by other researchers and producers The manuscript can be published, but subject to corrections 

General Comment: 

The use of propolis in medicine has a very good future due to its pharmacological action. Moreover, the production of propolis is environmentally friendly and can help to alleviate the current bee decline crisis. It also has socio-economic importance in countries such as Brazil, for example, providing benefits to people who are economically disadvantaged. The strong point of the manuscript is the detailed collection of data on the activity of propolis (antibacterial activity, antifungal activity, antiviral activity) and its synergistic activity with other antibacterial substances. Data on the use of this product as an adjunct in combating Covid-19 are also very promising. I propose to add more information on standardized propolis extracts (line 582). 

Specific Comments: 

Line 597. Please describe the composition of this standardized extract and what this standardization was about 

Line 599. Please describe the composition of this standardized extract and what this standardization was about 

Line 600. Please describe the composition of this standardized extract (GH 2002) and what this standardization was about 

Line 601. Please describe the composition of this standardized extract (Propoelix) and what this standardization was about 

Line 603. Please describe the composition of this standardized extract (M.E.D) and what this standardization was about 

Line 605. Describe the procedure in more detail. 

Line 606. Describe the procedure in more detail 

Line 609. Please describe the composition of this extract. 

Author Response

The author thank the overall positive appreciation of the paper by the Reviewer. The author thank the constructive comments and suggestions.

The modifications introduced, following the suggestion of the Reviewer, are outlined in yellow and are located between lines 599 and 626 of the revised version. Minor modifications were also introduced, not derived from suggestions of the Reviewer. They are outlined in grey.

Please see attachment, containing the revised version of the paper.
